# The Antimicrobial and Anti-Biofilm Effects of *Hypericum perforatum* Oil on Common Pathogens of Periodontitis: An In Vitro Study

Reza Bagheri [1,†], Sepideh Bohlouli [1,†], Solmaz Maleki Dizaj [2,*], Shahriar Shahi [2], Mohammad Yousef Memar [3,*] and Sara Salatin [2]

1 Department of Oral Medicine, Faculty of Dentistry, Tabriz University of Medical Sciences, Tabriz 51548-53431, Iran
2 Dental and Periodontal Research Center, Tabriz University of Medical Sciences, Tabriz 51548-53431, Iran
3 Infectious and Tropical Diseases Research Center, Tabriz University of Medical Sciences, Tabriz 51548-53431, Iran
* Correspondence: maleki.s.89@gmail.com (S.M.D.); yousef.memar64@gmail.com (M.Y.M.)
† These authors contributed equally to this work.

**Abstract:** The antibacterial and anti-biofilm effects of *Hypericum perforatum* oil against the common pathogens of periodontitis (*Escherichia coli*, *Streptococcus mutans*, *Staphylococcus aureus*, *Enterococcus faecalis*, *Porphyromonas gingivalis*) was investigated. Disk diffusion (DD), minimum inhibitory concentration (MIC), and minimum bactericidal concentration (MBC) approaches were applied to test the antimicrobial effects. In order to determine the anti-biofilm effects, the amount of bacterial biofilm formation was assessed using the microtiter plate technique. The anti-biofilm effects were then confirmed by determining the minimum biofilm inhibitor concentration (MBIC). The MIC, MBC, MBIC, and DD values were 64, 256, 512 μg/mL, and 14 mm for *Staphylococcus aureus*; 128, 256, 512 μg/mL, and 16 mm for *Streptococcus mutans*; 256, 512, 256 μg/mL, and 20 mm for *Escherichia coli*; 32, 128, 512 μg/mL, and 16 mm for *Enterococcus faecalis*; and 64, 128, 256 μg/mL, and 15 mm for *Porphyromonas gingivalis*, respectively. According to our results, *Hypericum perforatum* oil has antibacterial and anti-biofilm properties against the common bacteria associated with periodontitis.

**Keywords:** *Hypericum perforatum*; periodontitis; MIC; MBC; MBIC





## 1. Introduction

Periodontal diseases have impacted about 20–50% of the world's population [1]. Periodontitis is a chronic condition that begins with the existence of a biofilm that disturbs both the periodontal ligaments and the teeth [2]. The high occurrence of periodontal disease in all age ranges makes it a public health concern [1]. Periodontal disease is usually detected using objective assessments of the patient's medical history as well as clinical and radiographic checks [1]. Studies have shown that periodontal disease is related to quality of life, and that severe periodontitis has the greatest impact on quality of life by compromising functional and esthetic aspects [2]. Some factors, such as poor oral hygiene, smoking, diabetes, medication, age, heredity, and stress, are associated with periodontal diseases. Strong data has demonstrated the relationship between periodontal diseases and systemic diseases, such as cardiovascular diseases, diabetes, and adverse pregnancy outcomes [1]. Reducing the occurrence of periodontal disease may decrease the occurrence of these related systemic diseases and can also reduce their financial effects on health care systems [1,3].

Periodontal diseases are multifaceted and multi-microbial infectious situations. Human subgingival plaque has more than 500 bacterial species; however, large investigations have revealed that *Porphyromonas gingivalis* (*P. gingivalis*) is a significant etiologic

cause contributing to chronic periodontitis. It creates many virulence factors by modulating the host's inflammatory response, causing damage to the host's periodontal tissues [4]. Furthermore, numerous studies have revealed the importance of *P. gingivalis* [5–9], *Enterococcus faecalis* (*E. faecalis*) [10–13], *Escherichia coli* (*E. coli*) [14–20], *Streptococcus mutans* (*S. mutans*) [21–24], and *Staphylococcus aureus* (*S. aureus*) [25–31] bacteria in the development of periodontal diseases.

Conventional treatments, such as mechanical debridement (the use of ultrasonic and manual equipment) and also the application of local or systemic antibiotics, are the chief therapeutic methods in the treatment of chronic periodontitis [32]. While antibiotic use can diminish periodontal pathogens, it can also lead to bacterial resistance, allergies in patients, an inability to create the appropriate concentration of drugs in periodontal pockets, and some side effects [33,34]. Therefore, efforts to find alternative therapies have increased. One of these treatments involves antimicrobial substances of natural origin such as plants [35].

*Hypericum perforatum* (*H. perforatum*) is a medicinal plant in the Clusiaceae family, which is used in plant-based medicine for its antiseptic and antidepressant properties. In addition, it has been suggested to have antibacterial and antiviral effects and anti-inflammatory and analgesic action. *H. perforatum* oil contains flavonoids and phenolic acids, which demonstrate free radical scavenging activity. It exerts very effective anti-inflammatory effects in animal models with acute inflammation [36]. Numerous pharmacological actions are related to the presence of the hypericin and flavonoid constituents [37]. It has been reported that hypericin is responsible for the photosensitive reactions seen in this plant. Regarding the antidepressant effects of this plant, it seems that the other constituent of this plant, namely hyperforin, is the main antidepressant agent, and hypericin has a lesser role in these patients [37].

In this study, the antibacterial and anti-biofilm effects of *H. perforatum* oil were investigated on *S. aureus*, *S. mutans*, *E. coli*, *E. faecalis*, and *P. gingivalis*, all of which are the common pathogens of periodontitis. Our hypothesis was that *Hypericum perforatum* oil shows antimicrobial and anti-biofilm effects against these pathogens.

## 2. Materials and Methods

### 2.1. Materials

*H. perforatum* oil was produced by the Keroya Company. Brucella agar, vitamin K1, calcitonin antibiotic, defibrinated sheep blood, Thioglycolate broth, horse serum, trypsin reagent, amoxicillin, ciprofloxacin, metronidazole, amikacin, and gentamicin were purchased from Gibco, Ireland.

### 2.2. Disk Diffusion Method

The disk diffusion process was performed in order to assess the sensitivity of bacteria to *H. perforatum* oil. First, suspensions equivalent to 0.5 McFarland standard were prepared from the bacterial strains, and then a uniform lawn culture was grown on the surface of the Mueller Hinton agar for the *S. mutans*, *E. faecalis*, *E. coli*, and *S. aureus* bacteria using a sterile cotton swab. The *P. gingivalis* uniform lawn culture was performed on the surface of the Brucella agar, which was supplemented with sheep blood, hemin, and vitamin K. Next, sterile blank disks were immersed in *H. perforatum* oil (pure and 50% diluted with DMSO 0/1%; this amount of DMSO has no antibacterial effect) and the disks were placed on the agar surface at a certain distance from each other. Antibiotic disks of amoxicillin, metronidazole, ciprofloxacin, amikacin, and gentamicin were used as positive controls, and blank disks were used as the negative control. The plates were incubated for 24 h at 37 °C, after which the halos of non-growth around the disks containing antimicrobial substances were measured in millimeters from the back of the plate with a ruler.

### 2.3. Minimum Inhibitory Concentration (MIC)

The micro-broth dilution technique was performed in order to test the minimum inhibitory concentration of *H. perforatum* oil. In this way, for *S. mutans*, *E. faecalis*, *E. coli*, and

*S. aureus*, 100 mL of Mueller Hinton Broth medium was poured into the wells of a 96-well microplate, 100 mL of the *H. perforatum* oil was added to the first well of each row, and the dilution was performed in the next wells (one-half ratio). For *P. gingivalis*, the Brucella broth medium supplemented with lysed sheep blood, vitamin K, and hemin was used. In the other rows, ciprofloxacin, gentamicin, amoxicillin, amikacin, and metronidazole antibiotics were used as positive controls in the *E. faecalis*, *S. aureus*, *S. mutans*, *E. coli*, and *P. gingivalis* tests, respectively. Then, 100 mL of bacterial suspension ($10^5$ CFU/mL) was added to all the wells. The well containing water was considered a negative control. The wells were placed in a 35 °C incubator for 24 h and were then checked for turbidity caused by the growth of microorganisms. The microplate that was used for *P. gingivalis* was incubated in anaerobic conditions that were created by Gas-Pak in the jar. Methylene blue indicator was used to control the anaerobic conditions of the jar.

### 2.4. Minimum Bactericidal Concentration (MBC)

To perform this test for *S. mutans*, *E. faecalis*, *E. coli*, and *S. aureus*, the contents of the microplate wells in which the MIC determination test was performed (after checking the results) were transferred to the surface of the nutrient agar medium using a sampler, and then a uniform lawn culture was performed. The plates were then incubated at 35 °C for 24 h. After this time, the well with the lowest concentration of *H. perforatum* oil that led to no growth of bacteria on the surface of the nutrient agar plate was considered the MBC. Determination of the MBC for *P. gingivalis* was performed on the surface of the Brucella agar supplemented with sheep blood, vitamin K, and hemin under anaerobic conditions.

Ciprofloxacin, gentamicin, amoxicillin, amikacin, and metronidazole antibiotics were used as positive controls in the *E. faecalis*, *S. aureus*, *S. mutans*, *E. coli*, and *P. gingivalis* tests, respectively.

### 2.5. The Ability of Biofilm Formation

Before determining the anti-biofilm effects of *H. perforatum* oil, biofilm formation in the studied bacterial isolates was investigated. In order to determine the formation of biofilm in the bacterial isolates, a semi-quantitative method of biofilm determination was performed in 96-well microplates with flat ends. First, the suspension of fresh bacterial culture in TSB (Trypticase Soy Broth) medium was prepared with a 24 h incubation and its turbidity was adjusted to $10^7$ CFU/mL using optical density (OD) at a wavelength of 600 nm. Then, 100 mL of the bacterial suspension of each isolate was added to the end of each well of the 96-well microplate and was incubated at 37 °C for 48 h. After this time, the wells were washed with Phosphate-buffered saline X1 (pH: 7.4) and stained with 0.1% crystal violet for 30 min at room temperature. Then, the excess crystal violet was removed by washing and, after fixing with 95% ethanol, its OD was measured using a spectrophotometer at a wavelength of 570 nm. The well containing TSB medium without organism inoculation was used as a negative control [12]. This procedure was repeated three times for each of the investigated isolates and their average OD was calculated at a wavelength of 570 nm. The strains were classified as follows:

OD ≤ ODc = no biofilm formation (−)
ODc < ODt ≤ 2 × Odc = weak biofilm formation ability (+)
2 × ODc < ODt ≤ 4 × ODc = medium biofilm formation ability (++)
4 × ODt > ODc = strong biofilm formation ability (+++)

For *P. gingivalis*, all the mentioned steps were performed. However, the Brucella broth medium supplemented with lysed sheep blood was used and the microplate was incubated in anaerobic conditions.

### 2.6. Minimum Biofilm Inhibitory Concentration (MBIC)

In order to check the anti-biofilm properties of the studied oil, the minimum biofilm inhibitory concentration (MBIC) was used. For this purpose, after cultivation, a microbial suspension equivalent to 0.5 McFarland standard was prepared from the tested organisms

and cultured in 96-well polyester microplates containing Mueller Hinton broth medium. After incubation for 20 h at 37 °C, the contents of the wells were emptied, and the wells were washed with the sterile saline solution under sterile conditions. Then, serial concentrations of the studied oil were dispensed and incubated for 20 h at 37 °C. The contents of the wells were emptied again, and the wells were washed. Then, 200 mL of culture medium without the antimicrobial agent was added to each well. In the next step, the OD at 650 nm was read before and after 6 h of incubation, and the difference in this OD was compared with this difference in the positive control. MBIC was defined as the minimum concentration of antimicrobial substances that results in a difference below 10% compared to the standard OD. The well containing culture medium without antibiotics was examined for growth control and the well containing antibiotic and culture medium was examined for contamination control. The well containing the culture medium without the tested substances was used as a positive control in order to compare the ODs. For *P. gingivalis*, all the mentioned steps were performed. However, the Brucella broth medium supplemented with lysed sheep blood was used and the microplate was incubated in anaerobic conditions [12].

### 2.7. Time Kill Kinetics

After obtaining the MBC of the *H. perforatum* oil using the micro-broth dilution method, its antimicrobial effects were evaluated using the Time Kill Kinetics evaluation method. For this purpose, the initial amount of $1 \times 10^6$ CFU/mL of the desired bacteria was exposed to a concentration of *H. perforatum* oil that was equal to the MBC in each bacterial strain and was cultured in Mueller Hinton Broth medium (incubated at 35 °C). Then, at time intervals of 0, 1, 2, 3, 4, 6, 12, and 24 h, 0.1 mL of the microbial suspension cultured in the broth was transferred to the surface of the TSA medium and a uniform lawn culture was performed. After 24 h, the number of colonies was counted. The culture medium containing the organism without the presence of an antimicrobial substance was used as a control. A decrease in the amount of Log10 in CFU/mL of bacteria of at least three times was considered the bactericidal effect of the studied concentration. For *P. gingivalis*, all the mentioned steps were performed. However, the Brucella broth and Brucella agar medium supplemented with lysed sheep blood was used and the microplate was incubated in anaerobic conditions [38].

### 2.8. Statistical Analysis

The results were reported as descriptive indices. The Shapiro–Wilk test was used to check the normality of the units. In order to compare the MIC and MBC between the control and intervention groups, the Mann–Whitney test was used, and the Kruskal–Wallis test was used to compare the groups simultaneously. Furthermore, the Kruskal–Wallis test was used to compare the non-growth diameter zone and the ability to form biofilm, and the MBIC between the groups. SPSS version 25 software was used for the data analyses. A value of less than 0.05 was considered a significant level.

## 3. Results

### 3.1. Disk Diffusion

The median non-growth zone diameter for the *P. gingivalis* bacteria in pure oil, 50% diluted oil, and its positive control (metronidazole) was 15, 11, and 22 mm, respectively, and this difference was statistically significant ($p$-value = 0.026). Further analysis showed that a significant difference was seen only between 50% oil and metronidazole ($p$-value = 0.007). There was no significant difference between the 50% extract and the pure extract ($p$-value = 0.176) and between the pure oil and metronidazole ($p$-value = 0.176). This means that the pure oil has the same antibacterial effect as metronidazole against *P. gingivalis*.

The median non-growth diameter for the *S. mutans* microorganism in pure oil, 50% diluted oil, and its positive control (amoxicillin) was 16, 14, and 24 mm, respectively. This difference was statistically significant ($p$-value = 0.034). Further analysis

showed that a significant difference was seen only between the 50% extract and amoxicillin (*p*-value = 0.010) and there was no significant difference between the 50% extract and the pure extract (*p*-value = 0.286) and between the pure extract and amoxicillin (*p*-value = 0.128). This means that the pure oil has the same antibacterial effect as amoxicillin against *S. mutans*.

The median non-growth diameter for the *E. faecalis*, *S. aureus*, and *E. coli* bacteria in pure oil, 50% diluted oil, and their positive controls (ciprofloxacin, gentamicin, and amikacin, respectively) were 16, 15, and 16 mm (*p*-value = 0.325), 14, 14, and 18 mm (*p*-value = 0.054), and 20, 17, and 19 mm (*p*-value = 0.083), respectively. These differences were not statistically significant. This means that for the *E. faecalis*, *S. aureus*, and *E. coli* bacteria, the pure oil, 50% diluted oil, and their positive controls (ciprofloxacin, gentamicin, and amikacin, respectively) have similar antibacterial effects.

### 3.2. MIC Results

The median MIC values of oil and the antibiotics (control group) were 64 and 64 µg/mL in *S. aureus* (*p*-value = 0.7), 128 and 32 µg/mL in *S. mutans* (*p*-value = 0.1), 256 and 128 µg/mL in *E. coli* (*p*-value = 0.7), 32 and 16 µg/mL in *E. faecalis* (*p*-value = 0.2), and 64 and 64 µg/mL in *P. gingivalis* (*p*-value = 0.7), respectively. The highest and lowest MIC values for the *H. perforatum* oil were reported for *E. coli* and *E. faecalis*, respectively. This means that in *E. faecalis*, *S. aureus*, *S. mutans*, *E. coli*, and *P. gingivalis*, *H. perforatum* oil and the control group corresponding to each bacterium (ciprofloxacin, gentamicin, amoxicillin, amikacin, and metronidazole antibiotics, respectively) have similar inhibitory effects on the growth of the bacteria.

### 3.3. MBC Results

The median MBC values of the extract and the antibiotics (control group) were 256 and 128 µg/mL in *S. aureus* (*p*-value = 0.2), 256 and 64 µg/mL in *S. mutans* (*p*-value = 0.1), 512 and 256 µg/mL in *E. coli* (*p*-value = 0.1), 128 and 64 µg/mL in *E. faecalis* (*p*-value = 0.2), and 128 and 64 µg/mL in *P. gingivalis* (*p*-value = 0.2), respectively. The highest and lowest MBC values for the *H. perforatum* oil were reported for *E. coli* and *E. faecalis*, respectively. This means that in *E. faecalis*, *S. aureus*, *S. mutans*, *E. coli*, and *P. gingivalis*, *H. perforatum* oil and the control group corresponding to each bacterium (ciprofloxacin, gentamicin, amoxicillin, amikacin, and metronidazole antibiotics, respectively) have similar antibacterial effects. Table 1 shows the median non-growth zone diameters, MICs, and MBCs for the studied bacteria.

**Table 1.** The median non-growth zone diameters, MICs, and MBCs for the studied bacteria.

| Organism * | Non-Growth Zone Diameters (mm) | | | MIC (µg/mL) | | MBC (µg/mL) | |
|---|---|---|---|---|---|---|---|
| | **Pure Oil** | **50% Oil** | **Antibiotic** | ***H. perforatum* Oil** | **Antibiotic** | ***H. perforatum* Oil** | **Antibiotic** |
| *E. faecalis* | 15.67 ± 1.53 | 15 ± 1 | 16.33 ± 0.58 | 42.67 ± 18.48 | 21.33 ± 9.24 | 149.33 ± 97.76 | 64 ± 0 |
| *S. aureus* | 14.67 ± 1.15 | 13.67 ± 1.53 | 18.33 ± 0.58 | 85.33 ± 36.95 | 64 ± 0 | 213.33 ± 73.9 | 106.67 ± 36.95 |
| *S. mutans* | 15.67 ± 1.53 | 13.67 ± 0.58 | 24.33 ± 0.58 | 106.67 ± 36.95 | 37.33 ± 24.44 | 213.33 ± 73.9 | 53.33 ± 18.48 |
| *E. coli* | 19.67 ± 1.53 | 17.33 ± 0.58 | 19.67 ± 1.15 | 213.33 ± 73.9 | 170.67 ± 73.9 | 512 ± 0 | 213.33 ± 73.9 |
| *P. gingivalis* | 15 ± 1 | 11.33 ± 0.58 | 21.67 ± 0.58 | 53.33 ± 18.48 | 64 ± 0 | 256 ± 221.7 | 85.33 ± 36.95 |

* Ciprofloxacin, gentamicin, amoxicillin, amikacin, and metronidazole antibiotics were used as positive controls for the *E. faecalis*, *S. aureus*, *S. mutans*, *E. coli*, and *P. gingivalis* tests, respectively.

### 3.4. The Ability of Biofilm Formation Results

The mean OD values for *S. aureus*, *S. mutans*, *E. coli*, *E. faecalis*, *P. gingivalis*, and the negative control group were 0.912, 0.708, 0.643, 0.824, 0.571, and 0.184, respectively. Based on these values, *E. faecalis* and *S. aureus* had strong biofilm formation ability, and *S. mutans*, *E. coli*, and *P. gingivalis* had medium biofilm formation ability.

### 3.5. MBIC Results

The median MBIC values for *S. aureus*, *S. mutans*, *E. coli*, *E. faecalis*, and *P. gingivalis* were 512, 512, 256, 512, and 256 µg/mL, respectively.

### 3.6. Time Kill Kinetics

The results of the Time Kill Assay were 128 μg/mL, 256 μg/mL, 256 μg/mL, 512 μg/mL, and 128 μg/mL for *E. faecalis*, *S. aureus*, *S. mutans*, *E. coli*, and *P. gingivalis*, respectively. Figure 1a–e shows the results for the Time Kill Assay for the studied bacteria.

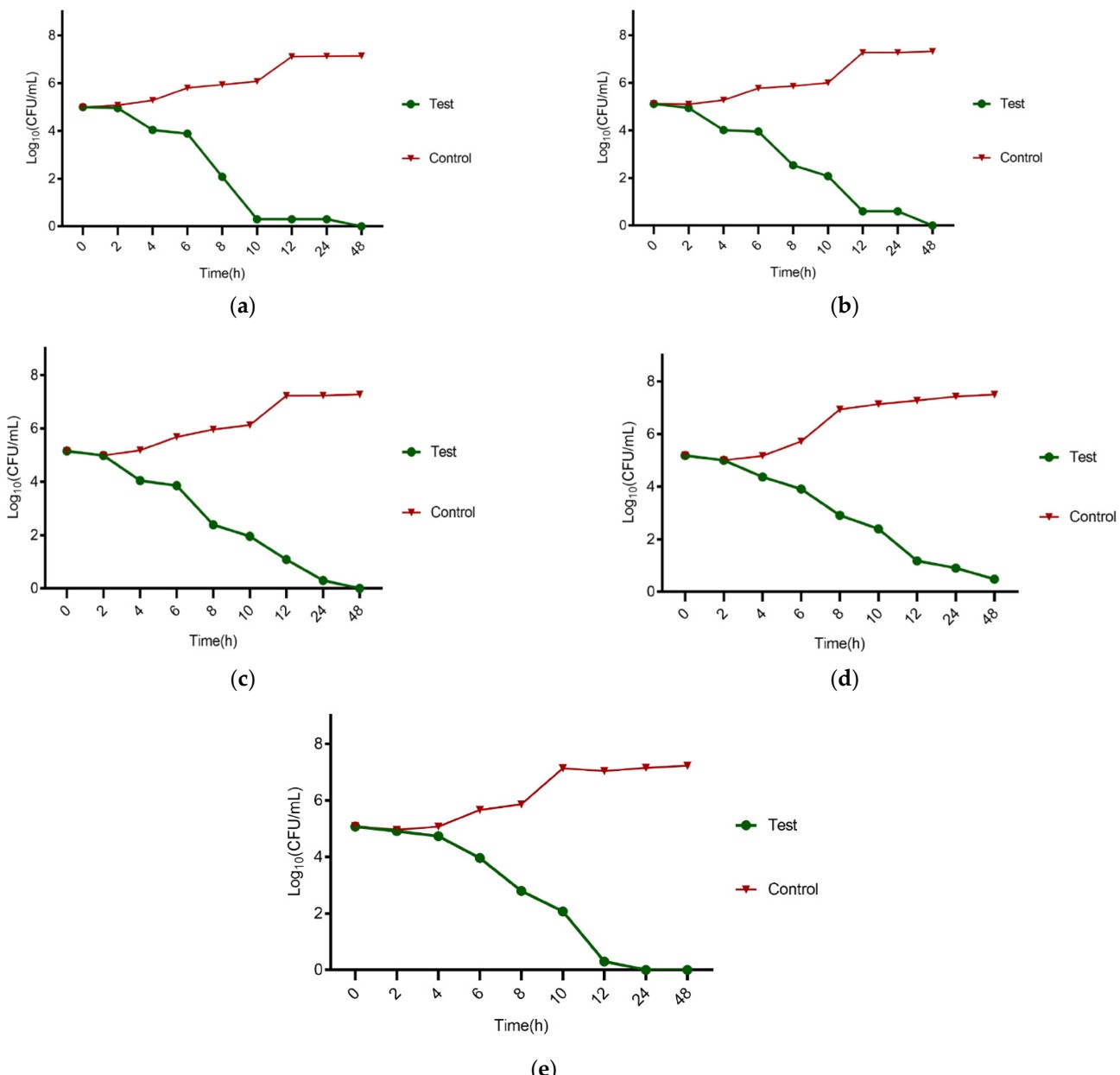

**Figure 1.** The results for the Time Kill Assay; (**a**) *E. faecalis*, (**b**) *S. aureus*, (**c**) *S. mutans*, (**d**) *E. coli*, and (**e**) *P. gingivalis*, respectively.

## 4. Discussion

Periodontal diseases are multifaceted and multi-microbial infections [4]. Various studies have shown the importance of *P. gingivalis* [4–9], *E. faecalis* [10–13], *E. coli* [14–20], *S. mutans* [21–24], and *S. aureus* [25–31] bacteria in the development and progression of periodontal diseases.

Biofilms, which are structured, surface-attached communities of bacteria, show a critical role in the expansion and pathogenesis of many infections because they are problematic to eliminate owing to their resistance to antimicrobials and host defence mechanisms [11,12].

A study conducted by Kerr et al. demonstrated that *P. gingivalis* can survive through several resistance factors, particularly biofilm formation via fimbriae, which are multifunctional adhesives, and signalling within bacterial biofilms and host tissues [39]. The study by Krzyściak et al. confirmed that *S. mutans* is capable of colonizing the oral cavity and can also form a bacterial biofilm through the mechanism of adhesion to a solid surface. Additional features that enable *S. mutans* to colonize the oral cavity include its ability to survive in an acidic environment and specific interactions with other colonizing microorganisms of this ecosystem [40]. In an in vitro study conducted by Thurnheer in 2015, *E. faecalis* and *E.coli* bacteria were shown to successfully colonize and grow in a biofilm composed of supragingival species [41]. A study conducted by Uribe-García et al. in 2021 showed that the expression percentage of virulence genes and genes involved in biofilm formation was higher in *S. aureus* strains that were isolated from patients with moderate periodontitis [27]. In this study, it was also shown that *E. faecalis* and *S. aureus* had strong biofilm formation ability, and that *S. mutans*, *E. coli*, and *P. gingivalis* had medium biofilm formation ability.

In the present study, when investigating the lowest concentration of *H. perforatum* oil at which the bacteria were eradicated, it was shown that concentrations of 128, 256, 256, 512, and 128 $\mu g \cdot mL^{-1}$ eradicates *E. faecalis*, *S. aureus*, *S. mutans*, *E. coli*, and *P. gingivalis*, respectively, in the culture medium.

In 2017, Khademnejad et al. proved the antimicrobial activity of *H. perforatum* against isolated samples of oral lactobacillus. According to their study, the hypericin present in this plant with an MIC of 0.625 removed acid-producing strains in the mouth and may be a suitable alternative for mouthwash and oral disinfectants [42]. Furthermore, in 1999, Schempp et al. investigated the antimicrobial activity of hyperforin. For the first time, they described the effect of hyperforin against multi-resistant strains of *S. aureus* with an MIC value of 1 $\mu g \cdot mL^{-1}$ [43]. The results of another study showed that treatment with *H. perforatum* reduced inflammation and tissue injury, and events related to periodontitis in Sprague–Dawley rats [44]. In another study, it was shown that a mixed hydroalcoholic extract of *H. perforatum* and *Calendula officinalis* may be considered an alternative treatment for periodontitis [45].

Yesilada et al showed that *H. perforatum* prevented the growth of five out of nine *Helicobacter pylori* strains with MIC values ranging from 15.6 μg/mL to 31.2 μg/mL. In comparison, the standard antibiotic, ofloxacin, showed MIC values between 0.49 μg/mL and 0.98 μg/mL [46]. Similarly, in the present study, it was shown that *H. perforatum* oil was effective against bacteria involved in periodontitis, such as *E. faecalis*, *S. aureus*, *S. mutans*, *E. coli*, and *P. gingivalis*, and when compared to the minimum inhibitory concentration of the *H. perforatum* oil between the study group and the control group (which only used antibiotics), our results showed that there was no significant difference between the two groups.

Regarding oral diseases, Vollmer et al. studied the effects of *H. perforatum* in combination with Visible Light Plus Water-Filtered Infrared-A (VIS + wIRA) to eradicate bacteria in oral biofilms. Based on their results, *H. perforatum* extract has a unique photo-activation potential at a concentration of 32 mg/mL. Finally, this plant caused the complete elimination of biofilms in the oral environment, and it was shown that this plant can be effective in the treatment of oral diseases that are associated with biofilms [47]. In our study, *H. perforatum* oil at concentrations of 512, 512, 256, 512, and 256 μg/mL inhibited the formation of biofilm in *S. aureus*, *S. mutans*, *E. coli*, *E. faecalis*, *and P. gingivalis*, respectively.

Furthermore, the antimicrobial properties of the extracts of this plant against bacteria and the formation of biofilm in the mouth were investigated by Süntar et al. in 2015. These researchers concluded that the aqueous extract of this plant had a strong antibacterial result in inhibiting the growth of *Streptococcus sobrinus* and *Lactobacillus plantarum*, and has a moderate antibacterial effect in inhibiting the growth of *S. mutans* and *E. faecalis* at 32 μg/mL and 16 μg/mL concentrations, respectively. Furthermore, it had anti-biofilm effects against *S. mutans* and *E. faecalis* at a concentration of 20.08 $\mu g \cdot mL^{-1}$, and can be used as a natural antibacterial agent in oral health products [48]. In this study, the minimum inhibitory concentration of *H. perforatum* oil for *S. mutans* and *E. faecalis* bacteria was reported as

128 and 32 μg·mL$^{-1}$, respectively, and the minimum inhibitory concentration of biofilm formation was reported as 512 μg·mL$^{-1}$ for these two species of bacteria. This difference could be due to the use of oil in this study instead of an aqueous extract of *H. perforatum*. The type, nature, and concentration of the chemical compounds differ between the aqueous extract and the oil. In addition, plants grown in different areas show different specific chemicals. Furthermore, the polarity of the solvent (methanol, ethanol, acetone, water, etc.) used in the extraction affects its composition [49].

Lasik et al. tested extracts of *H. perforatum* prepared with water and solutions of 10% and 30% water/ethanol (*v/v*) in terms of their antagonistic properties against *Enterococcus faecium*, *Bifidobacterium animalis*, *Lactobacillus plantarum*, and *E. coli* isolated from the human colon. The highest inhibitory effect was observed with the 30% ethanol solution [50]. In the present study, the pure extract and 50% oil extract of *H. perforatum* were used to determine its antibacterial effects using the disk diffusion method. No statistically significant difference was found between the pure extract and the 50% extract.

In two studies performed by Delcanale et al. and Bahmani et al., the antibacterial effect of *H. perforatum* extract was studied exclusively in a culture medium with *S. aureus*. Delcanale et al. found that if this bacterium was incubated with the hypericin present in the *H. perforatum* extract, this substance exerted photodynamic activity against *S. aureus* [51]. Bahmani et al. also reported that the hydroalcoholic extract of *H. perforatum* had antibacterial effects against *S. aureus* at an MIC equal to 625 and an MBC equal to 10,000 μg/mL [52]. Similarly, in our study, *H. perforatum* extract produced antibacterial effects at an MIC equal to 64 μg/mL and an MBC equal to 256 μg/mL against *S. aureus*, respectively.

In a similar study, Arpag et al. examined the influence of *H. perforatum* oil in preventing the growth of *P. gingivalis* bacteria and obtained an MIC equal to 0.312 μg/mL [53]. However, in the present study, this value was equal to 64 μg/mL. Nawchoo et al. investigated the effect of the methanol extract of *H. perforatum* found in endemic areas on various bacteria including *E. coli*. These researchers found that this substance had antibacterial effects at an MIC equal to 3.12 mg/mL [54]. In our study, the MIC against this bacterium was 256 μg/mL.

Although most of the common antibiotics work through the DNA destruction pathway [55], no proven antimicrobial mechanism has been reported in the scientific literature for *H. perforatum* oil.

## 5. The Strengths and Limitations

The results of this study showed that *H. perforatum* oil had appropriate antibacterial and anti-biofilm effects against the selected bacteria. This may be very helpful knowledge for overcoming bacterial resistance. In addition, the concentrations obtained in this study were lower compared to previous research works, which improves the hope of preparing optimal formulations based on this oil.

There are also other types of periodontal bacterial pathogens, such as *Fusobacterium nucleatum*, *Prevotella Intermedia*, and *Aggregatibacter Actinomicetencomitans*. *H. perforatum* oil should also be tested against these bacteria in future studies.

This study was an in vitro investigation. Thus, the toxicity of *H. perforatum* oil should be tested in future studies before any animal or clinical tests are conducted. In addition, the antimicrobial and anti-biofilm mechanisms for *H. perforatum* oil should be studied in order to verify its exact function.

## 6. Conclusions and Future Perspectives

According to our results, *H. perforatum* oil has antibacterial and anti-biofilm properties against the common bacteria associated with periodontitis. Furthermore, our study showed the same inhibitory and bactericidal effects between *H. perforatum* oil and the antibiotic control groups that were used for each of the bacteria in this study. Hence, we recommend conducting more studies in order to investigate the possibility of using this substance instead of antibiotics due to their limitations. Furthermore, further studies are required in

order to investigate the possibility of using this substance in oral health products and in the treatment of periodontitis. Newly developed biomaterials based on *H. perforatum* oil may be introduced and may have unique antimicrobial properties that can be utilized in other areas of dentistry, including tissue engineering and in wound dressings for exudative and long-term healing wounds.

**Author Contributions:** Methodology, S.B., S.S. (Shahriar Shahi) and M.Y.M.; Software, M.Y.M.; Formal analysis, R.B.; Investigation, M.Y.M.; Data curation, R.B. and S.S. (Sara Salatin); Writing—original draft, R.B. and S.B.; Writing—review & editing, S.M.D. All authors have read and agreed to the published version of the manuscript.

**Funding:** This study has been funded by the Tabriz University of Medical Sciences.

**Institutional Review Board Statement:** The ethics code was received from the Ethics Committee of Tabriz University of Medical Sciences, and all the steps were performed after obtaining the code (IR.TBZMED.VCR.REC.1400.384). All the participants gave written informed consent.

**Informed Consent Statement:** Not applicable.

**Data Availability Statement:** The raw/processed data required to reproduce these findings can be shared after publication by requesting them from the corresponding author.

**Acknowledgments:** This study was based on a thesis (No. 66911) registered at the Faculty of Dentistry, Tabriz University of Medical Sciences, Tabriz, Iran. We greatly acknowledge the Tabriz University of Medical Sciences, Tabriz, Iran for their financial support.

**Conflicts of Interest:** The authors state that there are no competing interest in this study.

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
