# Peer review of "The Antimicrobial and Anti-Biofilm Effects of Hypericum perforatum Oil on Common Pathogens of Periodontitis: An In Vitro Study"

_clinpract, doi:10.3390/clinpract12060104_

Round 1

Reviewer 1 Report

Comments:

The topic of the present original study, evaluating the in vitro antimicrobial and anti-biofilm effects of Hypericum perforatum oil on common pathogens of periodontitis is very interesting and innovative, especially considering antibiotic stewardship.

The issue seems particularly timely and has been widely described.

Reported findings currently presented may pave the way for further clinical investigations and may be clinically relevant in the future from the perspective of periodontal dysbiosis reversal and biofilm control in periodontal management.

The manuscript is well organized and written.

The introduction and Discussion sections need to be slightly expanded. Materials and Methods as well as the Results are clearly presented.

Concerns and suggestions:

Title:

·       I would suggest to add “: an in vitro study”.

Introduction:

·       Please, correct the period in lines 30-32 “1 The diagnosis of 30 periodontal disease is usually based on objective evaluations of the patient's medical/den- 31 tal history as well as clinical and radiographic examinations.”, according to “Staging and grading of periodontitis: Framework and proposal of a new classification and case definition - doi: 10.1002/JPER.18-0006”;

·       Please, add to the period in lines 34-35 “compromising functional and 34 esthetic aspects” …”and requiring complex prosthetic rehabilitations” “Computed tomography evaluation of jaw atrophies before and after surgical bone augmentation. Int J Clin Dent 2019;12(4):259-270 - eid=2-s2.0-85081018833”;

·       Please, add to period in lines 38-40 “Reducing the incidence and 38 prevalence of periodontal disease can reduce the systemic diseases associated with its and 39 can also minimize their financial impact on the health care systems” … “evidence-based recommendations on periodontal practice and the management of periodontal patients during and after the covid-19 era: challenging infectious diseases spread by airborne transmission. Open Dentistry Journal, 2021; 15. 325-336 - DOI: 10.2174/1874210602115010325”.

Materials and Methods section:

  • You tested “E. faecalis, S. aureus, S. mutans, E. coli, and P. 117 gingivalis”, and what about other periodontal pathogens, such as Fusobacterium nucleatum, Prevotella Intermedia, Aggregatibacter Actinomicetencomitans? Why were they not tested? Please, specify it within the text.

Discussion section:

  • strengths and limitations should be added (also highlighting that some suspected periodontal pathogens were not tested);
  • future perspectives should be added.

Author Response

Concerns and suggestions:

Title:

  • I would suggest to add “: an in vitro study”
  •  
  • Response: Thanks for your valuable comments. It has been done and highlighted in yellow.

Introduction:

  • Please, correct the period in lines 30-32 “1 The diagnosis of 30 periodontal disease is usually based on objective evaluations of the patient's medical/den- 31 tal history as well as clinical and radiographic examinations.”, according to “Staging and grading of periodontitis: Framework and proposal of a new classification and case definition - doi: 10.1002/JPER.18-0006”;
  • Please, add to the period in lines 34-35 “compromising functional and 34 esthetic aspects” …”and requiring complex prosthetic rehabilitations” “Computed tomography evaluation of jaw atrophies before and after surgical bone augmentation. Int J Clin Dent 2019;12(4):259-270 - eid=2-s2.0-85081018833”;
  • Please, add to period in lines 38-40 “Reducing the incidence and 38 prevalence of periodontal disease can reduce the systemic diseases associated with its and 39 can also minimize their financial impact on the health care systems” … “evidence-based recommendations on periodontal practice and the management of periodontal patients during and after the covid-19 era: challenging infectious diseases spread by airborne transmission. Open Dentistry Journal, 2021; 15. 325-336 - DOI: 10.2174/1874210602115010325”.

 Response: Thanks for your valuable comments. We paraphrased the mentioned texts and also added the related references. It has been done and highlighted in yellow.

Materials and Methods section:

  • You tested “E. faecalis, S. aureus, S. mutans, E. coli, and P. 117 gingivalis”, and what about other periodontal pathogens, such as Fusobacterium nucleatum, Prevotella Intermedia, Aggregatibacter Actinomicetencomitans? Why were they not tested? Please, specify it within the text.

Response: Thanks for your valuable comments. Yes, you are right. However, the mentioned bacteria was not in our access. We mentioned this point as a limitation of the study. It has been done and highlighted in yellow.

Discussion section:

  • strengths and limitations should be added (also highlighting that some suspected periodontal pathogens were not tested);
  • future perspectives should be added.
  • Response: Thanks for your valuable comments. It has been done and highlighted in yellow.

Reviewer 2 Report

Materials and Methods: all the methods used by the authors, very rigorously described, should have cited sources.

Results: some images:figures of the results would improve the content of the section.

Discussion:

Row 301 – an explanation regarding why there are different results in using oil vs aqueous extracts.

A phrase explaining why the authors used these specific antibiotics in the study should be added. What effects have the antibiotics at every level discussed in this study; how and if they are the same as those known about Hypericum perforatum – a parallel between their mechanisms could be incorporated in the discussion.

Although it is an in vitro study, an observation about the Hypericum perforatum toxicity or the lack of it should be added.

Author Response

Materials and Methods: all the methods used by the authors, very rigorously described, should have cited sources.

Response: Thanks a lot for your valuable comments. Some of them are routine methods like MIC and MBC. For the others, we added the references. It has been done and highlighted in yellow.

Results: some images: and figures of the results would improve the content of the section.

Response: Thanks a lot for your valuable comments. We added some.  It has been done and highlighted in yellow.

Discussion:

Row 301 – an explanation regarding why there are different results in using oil vs aqueous extracts.

Response: Thanks a lot for your valuable comments. We added some.  It has been done and highlighted in yellow.

A phrase explaining why the authors used these specific antibiotics in the study should be added. What effects have the antibiotics at every level discussed in this study; how and if they are the same as those known about Hypericum perforatum – a parallel between their mechanisms could be incorporated in the discussion.

Response: Thanks a lot for your valuable comments. We added some data in this regard to the discussion and limitation sections.  Please see. It has been done and highlighted in yellow.

Although it is an in vitro study, an observation about the Hypericum perforatum toxicity or the lack of it should be added.

Response: Thanks a lot for your valuable comments. We added some data in this regard to the limitation sections.  Please see. It has been done and highlighted in yellow.

Reviewer 3 Report

This is a basic study entitled "The antimicrobial and anti-biofilm effects of Hypericum perforatum oil on common pathogens of periodontitis. The idea is interesting for using antimicrobial substances of natural origin for the treatment of periodontitis. Minor concerns have been raised and should be resolved.

1.       One or two tables with values of disc diffusion, MIC, MBC, the ability of biofilm formation of the bacterial strains, MBIC, and Time Kill Kinetics assay are required.

2.       A great value will be added if the authors correlate the effect of Hypericum perforatum oil on different bacterial strains.

3.       Charts will also add a benefit for the manuscript and for the readers.

4.       Please add the hypothesis at the end of the introduction section.

5.       Minor English language polishing is also required.

Author Response

  1. One or two tables with values of disc diffusion, MIC, MBC, the ability of biofilm formation of the bacterial strains, MBIC, and Time Kill Kinetics assay are required.

Response: Thanks a lot for your valuable comments. We added some figures and tables. please see. It has been done and highlighted in yellow.

  1. A great value will be added if the authors correlate the effect of Hypericum perforatum oil on different bacterial strains.

Response: Thanks a lot for your valuable comments. We added some data in this regard. please see. It has been done and highlighted in yellow.

  1. Charts will also add a benefit for the manuscript and for the readers.

Response: Thanks a lot for your valuable comments. We added some figures and tables. please see. It has been done and highlighted in yellow.

  1. Please add the hypothesis at the end of the introduction section.

Response: Thanks a lot for your valuable comments.  It has been done and highlighted in yellow.

  1. Minor English language polishing is also required.

Response: Thanks a lot for your valuable comments. It has been done and highlighted in blue.

Round 2

Reviewer 2 Report

In my opinion, the manuscript has been improved.